# Investigation on the Thermal Characteristics of Enhancement-Mode p-GaN HEMT Device on Si Substrate Using Thermoreflectance Microscopy

**DOI:** 10.3390/mi13030466

**Published:** 2022-03-18

**Authors:** Hongyue Wang, Chao Yuan, Yajie Xin, Yijun Shi, Yaozong Zhong, Yun Huang, Guoguang Lu

**Affiliations:** 1Science and Technology on Reliability Physics and Application of Electronic Component Laboratory, China Electronic Product Reliability and Environmental Testing Research Institute, Guangzhou 510610, China; wanghongyue@pku.edu.cn (H.W.); 201811022423@std.uestc.edu.cn (Y.X.); huangyun@ceprei.com (Y.H.); luguog@126.com (G.L.); 2The Institute of Technological Sciences, Wuhan University, Wuhan 430072, China; chaoyuan@whu.edu.cn; 3Key Laboratory of Nano-Devices and Applications, Suzhou Institute of Nano-Tech and Nano-Bionics, Chinese Academy of Sciences (CAS), Suzhou 215123, China

**Keywords:** thermoreflectance, p-GaN HEMT, thermal characteristics

## Abstract

In this paper, thermoreflectance microscopy was used to measure the high spatial resolution temperature distribution of the p-GaN HEMT under high power density. The maximum temperature along the GaN channel was located at the drain-side gate edge region. It was found that the thermal resistance (*R*_th_) of the p-GaN HEMT device increased with the increase of channel temperature. The *R*_th_ dependence on the temperature was well approximated by a function of *R*_th_~*T*^a^ (a = 0.2). The three phonon Umklapp scattering, point mass defects and dislocations scattering mechanisms are suggested contributors to the heat transfer process for the p-GaN HEMT. The impact of bias conditions and gate length on the thermal characteristics of the device was investigated. The behaviour of temperature increasing in the time domain with 50 µs pulse width and different drain bias voltage was analysed. Finally, a field plate structure was demonstrated for improving the device thermal performance.

## 1. Introduction

The wideband gap semiconductor GaN has attracted significantly attention as a potential material for use in high-power, high-frequency and high temperature applications [1]. In recent years, the enhancement-mode p-GaN HEMTs on silicon substrate have been widely used in the power converters because of its superior performances [2,3]. The p-GaN HEMTs have been demonstrated to have ten times greater output power density (*P*) than that of the conventional Si devices due to the high field strength of GaN material and the polarization-effect-induced high-density high-mobility 2-D electron gas (2DEG) [4,5,6]. However, for the high-power density application, the junction temperature (*T*_J_) of the device can exceeds 150 °C, which has demonstrated detrimental consequences on the performance and reliability of the GaN devices [7,8]. Under the semi-on or saturation region operation, the hot electrons in the GaN channel emit many phonons to heat up the lattice, leading to a significant increase in device temperature, which is the well-known self-heating effects [9,10,11]. In addition to the huge heat generation, the decreased thermal conductivity of material with temperature increasing also limits the heat dissipation. Many reports have demonstrated that the thermal conductivity of the GaN films grown on substrates and the GaN bulk decrease with the increase of temperature because of the Umklapp and impurity scattering [12,13]. This temperature dependence effect is expected to impact the total thermal resistance (*R*_th_) of the p-GaN HEMT. The temperature dependence of the device *R*_th_ has been studied by 2D electrothermal simulation for the GaN devices grown on diamond [14]. However, experimental results are rare due to the challenging of probing the GaN channel temperature. Besides, the *R*_th_ of the device is linked not only to total power but also the bias conditions, which has been studied via micro-Raman thermography [15,16]. However, the Raman thermometry probe the temperature throughout the entire thickness of the GaN layer, resulting in the measurement of a through-thickness averaged temperature rather than that at the 2-DEG channel temperature [17]. Therefore, a non-invasive method that allows a direct measurement of the channel region without being skewed by through-thickness averaging is necessary to validate the thermal characteristics of the p-GaN HEMT, which is significant for verifying heat spreading simulations and obtaining thermal constrains on the device design.

In this work, accurate temperature values and distribution of the p-GaN HEMT under high power density are measured by thermoreflectance microscopy, which exhibits better spatial resolution and measurement efficiency than the Infrared reflectance and the Raman spectroscopy methodology, respectively [18,19]. The maximum temperature *T*_max_ along the GaN channel was located at the drain-side gate edge region. The *R*_th_ dependence on the bias conditions at the same power density is discussed. Besides, the thermal resistance of the p-GaN HEMT device dependence on the temperature was well approximated by a function of *R*_th_~*T*^a^ (a = 0.2). The three phonon Umklapp scattering, point mass defects and dislocation scattering mechanism are found to contribute to the heat transfer process for the p-GaN HEMT. The impact of gate length (*L*_G_) on the junction temperature of the device was investigated. The behaviour of temperature increasing in the time domain with 50 µs pulse width and different drain bias voltage (*V*_DS_) was analysed. Finally, a field plate structure was demonstrated for improving the device thermal characteristics.

## 2. Device Structure and Mechanism

The schematic device structure of the p-GaN HEMT studied in this work is shown in Figure 1. The epitaxy structure consists of silicon substrate, AlGaN transition layer, GaN buffer layer, AlGaN layer, p-GaN layer and passivation layer. The transfer and output curves of the device are shown in the Figure 2. The accurate temperature values and distributions along the GaN channel were obtained by the thermoreflectance microscopy system and the measurement scheme was based on the reflectivity (*R*) linear relation with the temperature (*T*): ΔR/R = *k*Δ*T*, where *k* is the thermoreflectance calibration coefficient. Detailed measurement methods can be found in the literatures [20,21]. The widely used techniques for evaluating the temperature of the transistors includes the infrared (IR) thermography, the Raman thermometry, and the thermoreflectance microscopy. Compared with the in IR thermography and Raman thermometry, the thermoreflectance microscopy used in this work exhibits higher spatial resolution, higher thermal mapping efficiency and can measure the temperature of metal more accurately [22]. The temperature in the GaN channel layer was measured by pulse laser of 365 nm wavelength, which is transparent for the SiN passivation layer and AlGaN barrier layer. The measured *k* at the access region of this device was ~9 × 10^4^ K^−1^.

## 3. Results and Discussion

The steady-state temperature distribution along the GaN channel of p-GaN HEMTs under *V*_DS_ = 90 V, *V*_GS_ = 6 V bias is shown in the Figure 3. It was found that the gate edge region of drain-side exhibited the highest temperature. It is well known that an electrical field peak is located at this region, where the electrons obtained a high velocity from the high electric field and exchange energy to the crystal, leading to a high crystal temperature. The accurate temperature values along the channel are shown in the Figure 3b. The temperature of the GaN channel under the gate metal region was not able to be measured because of the block of gate metal.

Average temperature in the gate-drain access region as a function of DC power (*I*_DS_
×
*V*_DS_) for the p-GaN HEMTs with different *V*_GS_ is shown in Figure 4a. The GaN channel temperature increased with the power density increase. It should be noted that the device shows different temperature when biased at the same power density with different *V*_GS_, which is also found in other literatures [15,23,24]. The device biased at *V*_GS_ = 6 V had higher current and lower *V*_DS_ under same power density, leading to a lower electric field strength and lower Joule heat. Meanwhile, for the device biased at *V*_GS_ = 6 V conditions, the electric field was rather uniformly spread out across the device channel compared to device biased at *V*_GS_ = 4 V case. Therefore, heat generation distribution was relatively uniform across the entire channel.

The temperature dependence on the *P* curves show an increasing slope trend. The increased temperature Δ*T* of the device can be expressed as Δ*T* = *R*_th_ × Δ*P.* Thus, the slope of the curves in the Figure 4a means the device’s *R*_th_ [14]. The derived *R*_th_ of the device with gate bias at 4 V and 6 V are shown in the Figure 4b. It was found that the *R*_th_ of the p-GaN HEMT increased with *T* increasing, which is commonly observed for the GaN bulk material [12,13]. For more qualitative analysis of heat transport in the device we consider thermal resistance *R*_th_ = $\kappa^{-1}$, where the κ is the thermal conductivity. In the first approximation, it can be expressed as [25,26],
(1)$$R_{\mathrm{th}}\;=\;AT^{-3}\;+\;BT^{-1}\;+\;CT^{-2}\;+\;DT\;+\;E\left(\frac{3T}{\theta_{D}}\right)\;\times \;\exp\left(-\frac{\theta_{D}}{3T}\right).$$

The consecutive terms of this formula are related to scattering of phonons on: (i) the sample boundaries, (ii) isolated dislocations, (iii) long-range strain fields (LRST) and two-dimensional imperfections, (iv) point mass defects. The last term describes the contribution of Umklapp scattering processes. The boundary scattering is usually dominant at low temperature, thus it was not considered in this work [27,28]. To evaluate the mechanism of the *R*_th_ dependence on the *T*, the experimental data were fitted by the equation:(2)$$R_{\mathrm{th}}\;=\;KT^{a}.\;\;\;\;\;\;\;\;$$

Results of fitting to the experimental data are shown in Figure 4b. The *K* = 7.3~8.6 and the index number *a* = 0.2 was obtained. It can be suggested from equation (1) that the index number should be 1 if the heat transport is dominated by the Umklapp process and point mass defects scattering. The small index number of 0.2 indicates that the dislocations also strongly affect the phonon transport for the p-GaN HEMT device. This is consistent with the high dislocation density of about 10^8^–10^11^ cm^−2^ observed in the GaN buffer layer grown on silicon substrate due to their large lattice mismatch [29,30,31]. Phonon can scatter on dislocations via two distinctive mechanisms. One is scattered by the elastic strain field surrounding the dislocation line, another is by the cores of the dislocation lines [32]. The exact underlying mechanism requires further in-depth consideration. Nevertheless, it should be noted that the dislocations can slow down the *R*_th_ degradation rate at high temperature [33,34].

Figure 5 shows the average temperature in the GaN channel dependence on the DC power for the device with different *L*_G_. It was found that the device with longer *L*_G_ exhibits lower *R*_th_. To explain this behaviour, the heat transport ways of the device are shown in Figure 5b. The heat generated in the GaN channel can both dissipate to the top gate metal and the substrate of the device. Therefore, the device with long *L*_G_ has a larger heat transport area in the top heat dissipation path, resulting in a lower *R*_th_.

The measured GaN channel average temperature transient curves with different *V*_DS_ under 50 µs pulse width and 10% duty cycle are shown in Figure 6. It was found that the device shows the same temperature rise and fall time at different *V*_DS,_ which is around 10 µs. The temperature transient can be written as:(3)$$\Delta T\left(t\right)\;=\;T_{max}\left(1\;-\;\exp\left(-\frac{t}{R_{th}C_{th}}\right)\right),\;\;\;\;\;\;\;\;\;$$
where the *t* is the time and the $C_{th}$ is the thermal capacitance. The extracted time constant $R_{th}C_{th}$ value of the p-GaN HEMT is 7 µs, which can be used in the device modelling and failure evaluating.

The self-heating of the device was characterized by the temperature transient measurement. The temperature maps at 0 µs, 6 µs, 10 µs and 23 µs with *V*_GS_ = 6 V and *V*_DS_ = 10 V are shown in Figure 7. It was found that the gate-source access region exhibits higher temperature than the values in the gate-drain access region and the temperature was uniform in the gate-drain access region. While for higher *V*_DS_ bias, the hottest region was on the edge of the gate near to the drain (Figure 8), which results from the extremely high electrical field strength at this region. For the p-GaN HEMT used as power switch, the devices normally operate in the linear region. The temperature of the device along the channel was uniform. When the device bias at high *V*_DS_, such as under the short-circuit stress, the temperature of the drain-side gate edge region increased dramatically within 20 µs.

Based on above analysis, a field plate structure was demonstrated for the device thermal characteristics improvement, which was simulated by the TCAD. The simulated device parameters are based on the fabricated p-GaN HEMT as shown in Figure 1. The gate-to-source length *L*_GS_, gate length *L*_G_ and gate-to-drain length *L*_GD_ are 4 µm, 6 µm and 10 µm. The SiN_x_ passivation, AlN layer, p-GaN layer, AlGaN layer, GaN channel, GaN buffer, and Si substrate were 300 nm, 20 nm, 60 nm, 12.5 nm, 50 nm, 2 um, and 500 µm, respectively. The ionized acceptor concentration and buffer trap density (*E*_V_ + 0.9 eV) were 3.5 × 10^17^ cm^−^^3^ and 3 × 10^16^ cm^−^^3^. The heat sink was at the bottom of the Si substrate and the temperature was set to 300 K. Figure 9 shows the lattice temperature distribution of the p-GaN HEMT device with and without source field plate structure. Benefiting from decreased heat generation at lower electrical field peak in the device with a filed plate, the peak temperature of the GaN channel decreased.

## 4. Conclusions

In summary, the temperature distribution of the p-GaN HEMT under high power bias was characterized by the thermoreflectance microscopy. It was found that the *R*_th_ of the p-GaN HEMT device increased with the increase of junction temperature, and their dependence was well approximated by a function of *R*_th_~*T*^a^ (a = 0.2). The large deviations from the traditional *R*_th_~T law resulted from the dislocation scattering effects on the phonons transport. The three phonon Umklapp scattering, point mass defects and dislocation scattering mechanism are suggested contributors to the heat transfer process for the p-GaN HEMT. The device biased at *V*_GS_ = 6 V exhibited lower average temperature along the channel compared to the device biased at *V*_GS_ = 4 V at the same power density. This indicates that the bias condition may have a relatively significant impact on device temperature and that this effect must be considered when building thermal models of devices under operation or undergoing accelerated life testing. Meanwhile, it was found that a long gate length was beneficial to heat dissipation. However, to improve the performance of the device, a short gate length is preferred in this field. There is a trade-off between the device electrical performance and temperature. The self-heating behaviour of the device was characterized by the temperature transient measurement with 50 µs pulse width and different *V*_DS_. The extracted time constant of the p-GaN HEMT was 7 µs. Finally, a field plate structure was demonstrated for improving the device thermal characteristics.

## Figures and Tables

**Figure 1 micromachines-13-00466-f001:**
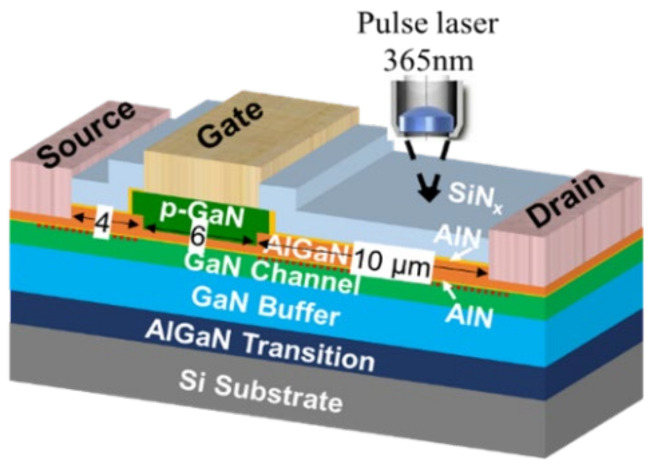
The schematic device structure of the p-GaN HEMT studied in this work.

**Figure 2 micromachines-13-00466-f002:**
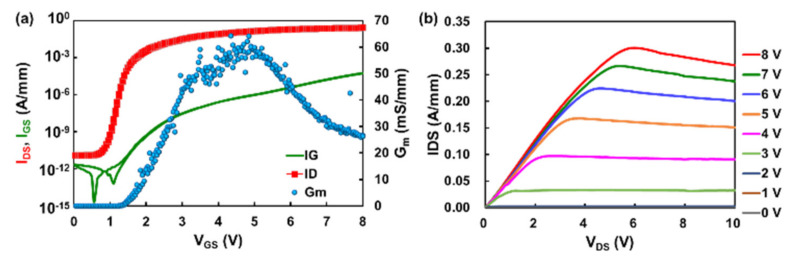
(**a**) Transfer and (**b**) output characteristics of the p-GaN HEMT with *L*_GS_/*L*_G_/*L*_GD_ = 4/6/10 µm.

**Figure 3 micromachines-13-00466-f003:**
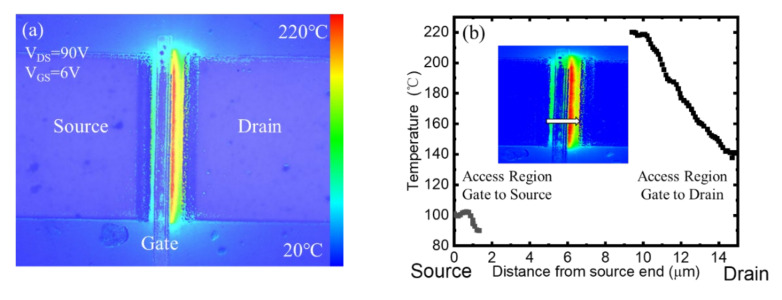
(**a**) Steady-state thermoreflectance maps of p-GaN HEMTs measured at 365 nm. (**b**) Temperature profiles of access region from the source to the drain.

**Figure 4 micromachines-13-00466-f004:**
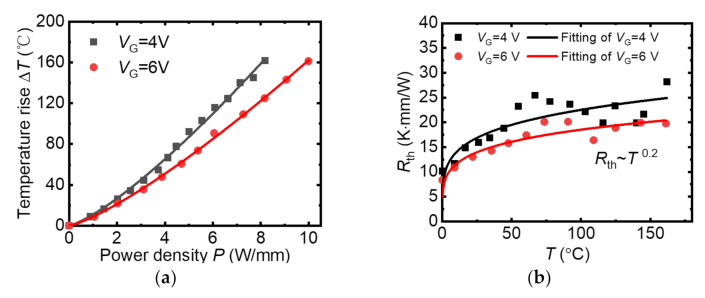
(**a**) Average temperature in the gate-drain access region as a function of DC power (*I*_DS_ × *V*_DS_) for the p-GaN HEMTs with different *V*_GS_ (**b**) The *R*_th_ dependence on the device temperature.

**Figure 5 micromachines-13-00466-f005:**
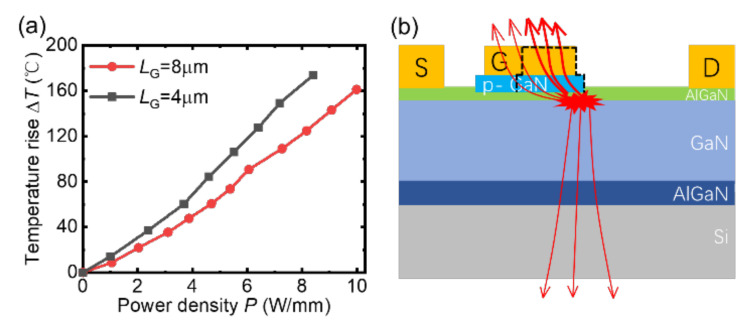
(**a**) Average temperature in the gate-drain access region as a function of DC power for the p-GaN HEMTs with different *L*_G_. (**b**) The heat transport ways of the p-GaN HEMT.

**Figure 6 micromachines-13-00466-f006:**
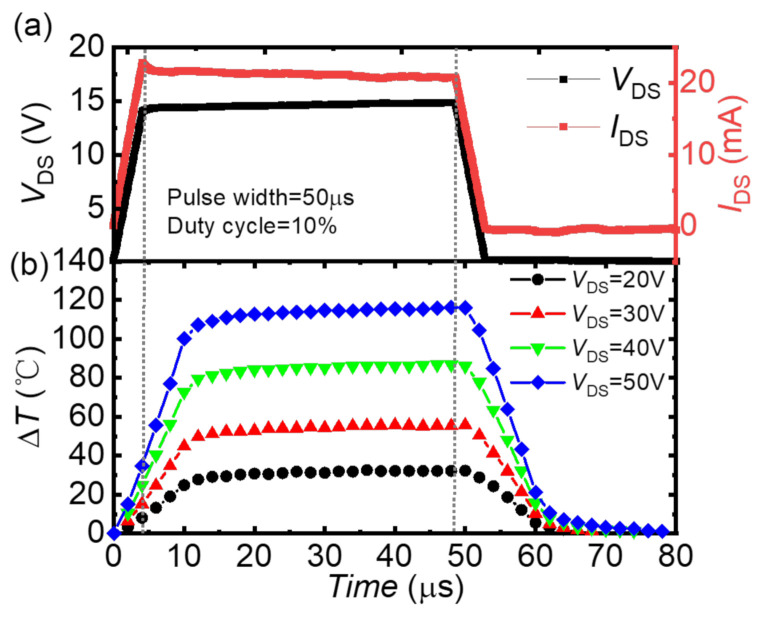
(**a**) The voltage and current curves under transient thermal measurement. (**b**) Transient temperature curves with different *V*_DS_.

**Figure 7 micromachines-13-00466-f007:**
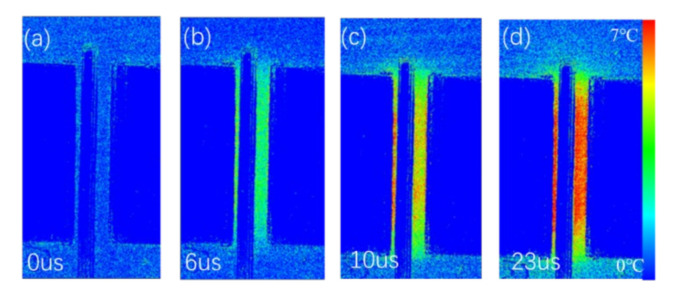
The measured temperature maps at (**a**) 0 µs, (**b**) 6 µs, (**c**) 10 µs and (**d**) 23 µs with *V*_GS_ = 6 V and *V*_DS_ = 10 V.

**Figure 8 micromachines-13-00466-f008:**
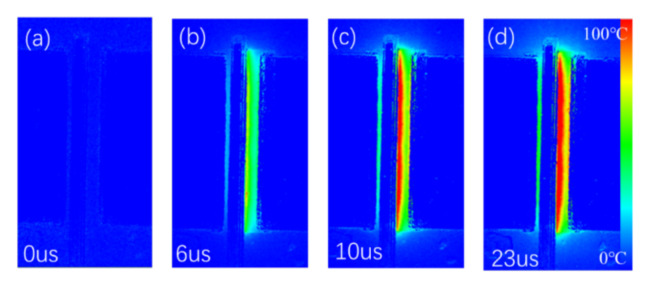
The measured temperature maps at (**a**) 0 µs, (**b**) 6 µs, (**c**) 10 µs and (**d**) 23 µs with *V*_GS_ = 6 V and *V*_DS_ = 50 V.

**Figure 9 micromachines-13-00466-f009:**
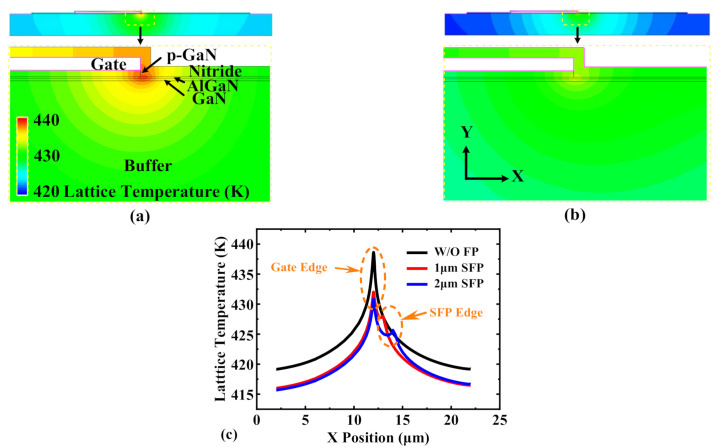
The simulated lattice temperature at *V*_GS_ = 4 V and *V*_DS_ = 60 V. P-GaN HEMT (**a**) without field plate, (**b**) with 1 µm source field plate (SFP), and (**c**) the lattice temperature along the GaN channel.

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
