# Peer review of "Investigation on the Thermal Characteristics of Enhancement-Mode p-GaN HEMT Device on Si Substrate Using Thermoreflectance Microscopy"

_micromachines, 2022, doi:10.3390/mi13030466_

Round 1

Reviewer 1 Report

The temperature distribution of p-GaN HEMT has been systematically studied. A lot of data has been presented in the manuscript, but few conclusion is addressed based on the data. Besides, I have some questions about the manuscript.

  1. The author mentioned “self-heating effects” and “the thermal conductivity of GaN material decrease with the increase of temperature”. Is there any relationship between these two?
  2. The motivation of this work is “the thermal characteristics of the p-GaN HEMT device on silicon substrate are rarely investigated”. The author should explain more about the relationship between this and abovementioned “self-heating effects” and “thermal conductivity of the GaN”.
  3. The author claimed “the three phonon Umklapp scattering, point mass defects and dislocations scattering mechanisms are found contributing to the heat transfer process for the p-GaN HEMT”, can they do some quantitative analysis of the contribution?
  4. In page 3, line 105, how did the author obtain the value of K and a?
  5. The author should explain more about the simulation part.
  6. The description of the manuscript requires careful polishing.

Reviewer 2 Report

Dear authors,
This article is acceptable after the following considerations.

1. In this manuscript, it is better to compare the method used to evaluate the temperature of the transistor with other conventional methods and determine the advantages of this method over the others.

2. It is mentioned in the conclusion section that increasing the length of the gate will cause more heat transfer to the environment outside the transistor. It should be noted that with the advancement of technology and reducing the size of transistors, researchers in this field want to reduce the gate length of transistors to increase the speed and efficiency of transistors.

3.The number of references used in the whole text is after the dot, which is better to come before the dot.

Best regards,

Round 2

Reviewer 1 Report

Authors has addressed all my questions.

Author Response

Thank you for the kind review.
